# Transcriptional control of a stem cell factor nucleostemin in liver regeneration and aging

Xiaoqin Liu[1,2☉], Junying Wang[1☉], Fang Li[1], Nikolai Timchenko[3], Robert Y. L. Tsai[1,2]*

**1** Institute of Biosciences and Technology, Texas A&M Health, Houston, TX, United States of America, **2** Department of Translational Medical Sciences, Texas A&M University School of Medicine, Bryan, TX, United States of America, **3** Department of Surgery, Cincinnati Children Hospital Medical Center, University of Cincinnati, Cincinnati, OH, United States of America

☉ These authors contributed equally to this work.
* robtsai@tamu.edu

**Data Availability Statement:** All relevant data are within the manuscript.

**Funding:** This work was supported by National Institute on Aging (NIA) Public Health Service

## Abstract

Nucleostemin (NS) plays a role in liver regeneration, and aging reduces its expression in the baseline and regenerating livers following 70% partial hepatectomy (PHx). Here we interrogate the mechanism controlling NS expression during liver regeneration and aging. The NS promoter was analyzed by TRANSFAC. Functional studies were performed using cell-based luciferase assay, endogenous NS expression in Hep3B cells, mouse livers with a gain-of-function mutation of C/EBPα (S193D), and mouse livers with C/EBPα knockdown. We found a CAAT box with four C/EBPα binding sites (-1216 to -735) and a GC box with consensus binding sites for c-Myc, E2F1, and p300-associated protein complex (-633 to -1). Age-related changes in NS expression correlated positively with the expression of c-Myc, E2F1, and p300, and negatively with that of C/EBPα and C/EBPβ. PHx upregulated NS expression at 1d, coinciding with an increase in E2F1 and a decrease in C/EBPα. C/EBPα bound to the consensus sequences found in the NS promoter *in vitro* and *in vivo*, inhibited its transactivational activity in a binding site-dependent manner, and decreased the expression of endogenous NS in Hep3B cells. *In vivo* activation of C/EBPα by the S193D mutation resulted in a 4th-day post-PHx reduction of NS, a feature shared by 16-m/o livers. Finally, C/EBPα knockdown increased its expression in aged (24-m/o) livers under both baseline and regeneration conditions. This study reports the C/EBPα suppression of NS expression in aged livers, providing a new perspective on the mechanistic orchestration of tissue homeostasis in aging.

## Introduction

Aging has a significant impact on the structure and function of liver [1–10]. One of the clinically relevant age-dependent changes in the liver is its reduced ability to regenerate with age, as shown by fewer hepatocytes reentering the S-phase after partial hepatectomy (PHx) (30% vs. 99%) and those that replicate do so less rapidly in the aged than in the young [2]. Consequently, the restored liver mass after injury is reduced and delayed in the elderly compared to

(PHS) grant R21AG052006 and the Cancer Prevention Research Institute of Texas (CPRIT) Individual Investigator Research Award (RP200081) to RYT. The funders had no role in study design, data collection and analysis, decision to publish, or preparation of the manuscript.

**Competing interests:** he authors have declared that no competing interests exist.

the young, creating health issues such as slower recovery from liver resection, decreased survival of transplanted livers obtained from older donors or grafted into older recipients, and a 3-to-5-fold increase in liver disease mortality [11–14].

Several mechanisms have been proposed to explain the effect of aging on liver regeneration, including decreased hepatic sensitivity to growth factors [15,16], increased reactive oxygen species, shortened telomere length [17,18], increased inflammatory response, accelerated cellular senescence, and reduced pseudocapillarization. At the molecular level, aging has been shown to increase the activity of CCAAT/enhancer-binding protein (C/EBP) and E2F4-Rb complex, which silences the expression of hepatocyte proliferative factors such as DNA polymerase α, c-Myc, cdc2, Forkhead Box protein M1, and p300 [5,6,19–23]. During aging, more genomic errors may accumulate due to insufficient repair of DNA damage caused by toxic agents from external sources or DNA replication *per se*. A recent genome-wide study revealed new differentially methylated regions and gene targets that are inversely affected by liver aging and regeneration, which may explain some of the adverse effects of aging on liver regeneration [24].

Nucleostemin (NS) was first discovered in neural stem cells [25,26] and was later found in multiple types of stem cells and cancers [27–30]. It belongs to a family of three GTP-binding nucleolar proteins [31]. NS expression is relatively low in most adult tissues but is upregulated during regeneration [32–35]. In the liver, NS plays a role in protecting regenerating hepatocytes from replicative DNA damage [35]. Mechanistically, loss of NS increases replicative DNA damage and impairs RAD51 recruitment to stalled replication-induced foci in stem/progenitor cells [36–38]. It has been shown that NS modulates the outcome of hepatocellular carcinoma (HCC) by helping tumor cells cope with genomic damage caused by replicative stress [39]. NS activity has also been linked to the activation of signal transducer and activator of transcription 3 (STAT3) in HCC [40,41], and STAT3 is known to be activated during liver regeneration [42]. These data indicate that NS may provide a mechanism that promotes hepatocyte proliferation by protecting the genome from replicative DNA damage during regeneration and oncogenesis.

Given the important role of NS in liver regeneration [35], we investigated the upstream mechanism that regulated its transcriptional expression during liver regeneration and aging. Our results showed that aging reduced NS expression in the liver both at baseline and during regeneration. Promoter and expression analyses indicated that NS expression was negatively regulated by C/EBPα during liver aging and regeneration. Further studies confirmed that C/EBPα bound NS promoter and suppressed its transcriptional activity, and depletion of C/EBPα in aged livers increased endogenous NS expression under both baseline and regenerative conditions. Using NS as a case in point, our study provides a new molecular perspective on the homeostatic regulation of liver regeneration during the aging process.

## Materials and methods

### Animal care and partial hepatectomy

Mice were housed in a vivarium with the light-dark cycle set at 12h-12h with lights on at 06:00 and off at 18:00, and allowed free access to normal chow diet and water. All procedures were in accordance with the *Guide for the Care and Use of Laboratory Animals* and approved by the Institutional Animal Care and Use Committee (2018-0396-IBT and 2021-0264-IBT). Animals were sacrificed at specific experimental endpoints as described in individual studies (*i.e.*, 2, 16, and 24 months old). Death was not a likely nor a planned endpoint of this study. No animal required premature euthanasia. 30–70% partial hepatectomy was performed based on a previously published procedure [35]. Wild-type (C57BL/6J) female mice were anesthetized by isoflurane inhalation. After laparotomy, cuts were made on the falciform ligament and

membrane between the caudate and left lateral lobes of the liver. For 70% (2/3) PHx, the left lateral lobe was resected, followed by resection of the right medial and left medial lobes. To perform 30% (1/3) PHx, which was used on 24-month-old mice for better post-operative survival, only the left lateral lobe was ligated with a 5–0 silk suture and resected. The peritoneum was closed using absorbable 4–0 suture. The skin was closed with an Autoclip wound clip (Becton Dickinson). To collect sham-operated samples, laparotomy and suture were performed without liver resection, and animals were euthanized by CO2 asphyxiation for tissue collection.

## Real-time RT-PCR assay

The $\Delta C(t)$ values between the target and reference genes were determined using the MyiQ single-color real-time PCR detection system and supermix SYBR green reagent, as described previously [43,44]. $\Delta\Delta C(t)$ values were calculated based on measurements of to 4–8 biological replicates with to 2–3 technical repeats for each biological sample. Primers were designed to set the Tm at 60°C for all reactions. The results were compared with those of two reference genes, Rps3 and Hmgn1. The sequences for mouse qRT-PCR primers are as follows: Ns, 5'-GT CTGATCTAGTACCAAAGG-3' and 5'-GGGAAACCAATCACTCCAAC-3'; c-Myc, 5'-CAGCGAC TCTGAAGAAGAGCAAG-3' and 5'-TAGTTGTGCTGGTGAGTGGAGAC-3'; E2f1, 5'-CTGGATCA CCTGATGCACATCTG-3' and 5'-TCCACAGCTTGTAGTTGGGTCTC-3'; p300, 5'-GCATGCGG TCTGTGAACAACATG-3' and 5'-TCCAGTACTAGATGGCTGAGCTG-3'; C/EBPα, 5'-CAAGAA GTCGGTGGACAAGAAC-3' and 5'-GGTCATTGTCACTGGTCAACTC-3'; C/EBPβ, 5'-AAGAA GACGGTGGACAAGCTGA-3' and 5'-TGCTTGAACAAGTTCCGCAGGG-3'; Rps3, 5'-ATGGCGG TGCAGATTTCCAA-3' and 5'-CATTCTGTGTCCTGGTGGC-3'; Hmgn1, 5'-GGGAAAGGAT AAAGCATCAGAC-3' and 5'-TTCAGAGGCTGGACTCTGGTT-3'. The sequences for human qRT-PCR primers are as follows: NS, 5'-GAACAAAGCCAAGTCGGG-3' and 5'-GTCCACTC TGGACAATGG-3'; c-MYC, 5'-AGCGACTCTGAGGAGGAACAAG-3' and 5'-TGCGTAGTTGT GCTGATGTGTG-3'; E2F1, 5'-CTGGACCACCTGATGAATATCTG-3' and 5'-TCTGAAAGT TCTCCGAAGAGTCC-3'; p300, 5'-CCATGAGCAACATGAGTGCTAG-3' and 5'-TCCAGTAGTGG ATGGTTGAGCTG-3'; C/EBPα, 5'-CAAGAAGTCGGTGGACAAGAAC-3' and 5'-GGTCATTGTCA CTGGTCAGCTC-3'; C/EBPβ, 5'-AAGAAGACCGTGGACAAGCACA-3' and 5'-TGCTTGAACA AGTTCCGCAGGG-3'; RPS3, 5'-CTTTCCAGAGGGCAGTGTAGA-3' and 5'-ATGAACCGCAGCA CACCATAG-3'; HMGN1, 5'-GGCAGCAGCGAAGGATAAATC-3' and 5'-TTCATCAGAGGCTGGA CTCTC-3'.

## Immunohistochemistry (IHC)

Tissue collection, processing, and immunostaining of NS followed the same procedures as described previously [41]. Sections were treated with 0.3% $H_2O_2$ and antigen-retrieved in boiling 10mM Tris solution (with 1mM EDTA pH9). IHC was performed by incubation with primary anti-rat NS antibody (Ab2438, 1/100×), followed by biotin-labeled secondary antibodies, avidin-conjugated HRP, and DAB color reaction. The specificity of Ab2438 was previously validated [45].

## Cell-based dual-luciferase reporter assay

Hep3B cells were purchased from ATCC (#HB-8064) and tested negative for mycoplasma infection on 6/29/20 and 3/29/24, respectively. Cells were grown in EMEM supplemented with 10% FBS and passaged every 2–4 days up to 10 passages after thawing. Cell-based dual-luciferase reporter assays were performed as described previously with some modifications [46]. Hep3B cells were seeded in 24-well plates and transiently transfected with Firefly (100ng) and

Renilla (10ng) reporter plasmids with or without C/EBPα (Addgene #66978) or C/EBPβ (Addgene #49198) expression plasmid (100ng or 300ng) using X-tremeGENE 9 DNA transfection reagent (Roche Cat# 06365787001). The total DNA amount in each well was adjusted to 410ng using the empty expression vector (pCIS). Whole-cell extracts were harvested after 24h of transfection plus 24h of recovery. Firefly and Renilla luciferase activities were measured using the Dual-Luciferase Reporter Assay System (Promega Cat# E1910) on a Cytation 5 Imaging Reader (BioTek). Data represent four biological replicates with one or two technical repeats each (n = 7).

## EMSA

Recombinant proteins were expressed, harvested in extraction buffer containing phosphate-buffered saline (PBS) without $Ca^{2+}$ or $Mg^{2+}$ ions, mixed with specified amounts of probes, and incubated on ice. The binding reaction mixture contains 0.2–0.4ng of probe with or without a 100-fold molar excess of unlabeled oligo-nucleotides, 10μg of nuclear extract or 30–40μg of whole cell extract, 2μg of poly(dI-dC), 20mM Tris-HCl (pH 7.6), 100mM KCl, 5mM $MgCl_2$, 1mM DTT, and 10% glycerol. The reaction products were subjected to electrophoresis on a 5% polyacrylamide gel (30:1.5) in 0.5× Tris-borate-EDTA (TBE) buffer at 4°C and detected by autoradiography. To generate EMSA probes, oligonucleotides were radiolabeled with γ-$P^{32}$ ATP in a T4 kinase reaction, annealed with excess amounts of antisense oligonucleotides, and purified using a nucleotide removal kit. Supershift experiments were performed by adding an anti-C/EBPα antibody to the reaction solution for 30 min.

## Chromatin Immunoprecipitation (ChIP)

Hepa1-6 cells (CRL-1830) were purchased and mycoplasma-tested from ATCC (3/29/22). Cells were grown in DMEM with high glucose (ATCC #30–2002), supplemented with 10% FBS and passaged every 2–4 days. Transfection of FLAG-tagged C/EBPα (Addgene #66978) was performed by using 10μg of plasmid DNA complexed with 15μL of jetPRIME reagent (Polyplus #101000046, New York, NY). After 48 hours, cells (4xE6) were harvested and cross-linked in 1% formaldehyde. Chromosomal DNAs were sheared to 200-500bp fragment by sonication, carried out in 200μL of shearing buffer for 2 minutes using the COVARIS M220 ultrasonic apparatus (Woburn, MA) with average incident power (AIP) of 7.5 watts t. 1/30th of the lysate was used to measure the input DNA amount. Immunoprecipitation was performed using the ChIP-IT Express Chromatin Immunoprecipitation Kit (Active Motif #53008, Carlsbad, CA) with 100μL of sheared chromatin sample, 3μg of anti-Flag antibody (Sigma #1804) or IgG (Sigma #I5381, Saint Louis, MO), and Protein G-magnetic beads. Immunoprecipitated DNAs were extracted by RNase-A and proteinase-K treatment and reverse crosslinking and subjected to qPCR assay. Each group consisted of four biological replicates and two technical repeats (n = 8). The qPCR primer sequences are as follows: Ns-p1, 5'-AATCCCAAACATGTCTTCAGCAG-3' and 5'-TTCCAGATGTTGAGGAACATAGC-3'; NS-p3, 5'-TTAGTCCTCGAATTCCTACATCC-3' and 5'-TGTACCGTCACGACGAGTTGAG-3'.

## shRNA design and production

Sense and antisense shRNA oligonucleotides were designed to target two different regions of 21 nucleotides within the coding region of mouse C/EBPα (shCEBPA1 and 2), or a scrambled sequence of 21 nucleotides as a negative control (shScr). The hairpin structure includes a stem-loop sequence of TTCAAGAGA, a 5' blunt end, and a 3' XhoI-compatible cohesive end. The target sense sequences were 5'-TCTCGCTTGGGCGAGAGTAAG-3' (shScr), 5'-AGAAGTCGGTGGACAAGAACA-3' (shCEBPA1), and 5'-GGAGTTGACCAGTGACAATGA-3' (shCEBPA2).

Each pair of oligonucleotides was ligated into the AAV gene transfer vector AV5-siRNA-GFP (Addgene #124972) at the HpaI and XhoI cloning sites. Purified plasmids were co-transfected with XX, Rep/Cap serotype 8, and AdΔF6 helper plasmids into 293T cells using the iMFectin transfection reagent (GenDepot). The cell viral lysate AAV and media-secreted AAV were combined and purified on a discontinuous iodixanol gradient and concentrated in Amicon filter centrifugation units (100,000 MW). AAV8 titers were quantified by absolute endpoint qPCR using WPRE-172 and WPRE-392 primers.

### Portal vein injection

C57BL6/J mice were anesthetized by isoflurane inhalation (2–4%) mixed with oxygen. Laparotomy was performed with a midline incision. Intestines were pulled out and covered with wetted gauze. AAV8 virus was injected at $1\times10^{11}$ genome copies (gc) in 50μL per animal via the portal vein using a 32G sterile needle with 3-5mm in depth and less than 5 degrees.

### Statistical analyses

Statistical analyses of qRT-PCR data comparing the expression of genes between two groups and the liver-to-body weight ratios between 2 and 16-m/o livers following PHx were performed using t-test. Relationship between the expression of NS and C/EBPα was calculation by Pearson's product-moment correlation using the cor.test() function in R 4.4.1.

## Results

### Expression profiles of NS in liver aging and regeneration

To investigate how NS expression was regulated during liver aging and regeneration, we measured NS expression in young (2-m/o), adult (8-m/o), and old (16-m/o) mice using qRT-PCR, as its baseline expression was low in adult livers. The results showed that NS expression was reduced by 40% in 8-m/o and 16-m/o livers compared to that in 2-m/o livers (Fig 1A1). In 2-m/o livers, NS expression rose rapidly by 35% at 1d following 70% PHx-induced regeneration, continued to increase at 2d and 4d, and returned to the baseline level at 7d (Fig 1A2). Compared to non-operated livers, NS expression showed no significant changes in the livers collected from 2-m/o mice at 1, 2, 4, and 7 days after receiving sham operation (*i.e.*, laparotomy) (Fig 1A3). At 16 m/o, the NS expression level also increased rapidly at 1d post-PHx but returned to the baseline level at 4d post-PHx, earlier than that at 2 m/o (7d) (Fig 1A4). To determine whether the increased expression of NS in liver regeneration could be detected at the protein level, anti-NS immunohistochemistry (IHC) was performed on liver samples collected at 0d and 2d post-PHx from 2-m/o and 16-m/o mice. The IHC results confirmed the increase of NS signals in the nucleolus at PHx-2d compared to PHx-0d at both 2 m/o (Fig 1B1) and 16 m/o (Fig 1B2). The increase of NS signals was manifested in both the signal intensity and the nucleolar size. No clear difference between the basal NS levels of the 2- and 16-m/o livers was detected by IHC. The increase in liver-to-body weight ratios at 1d, 2d, 4d, and 7d post-PHx (70%) were delayed in 16-m/o mice compared to that in 2-m/o mice (S1 Fig). These results demonstrate that aging attenuates the baseline expression level of NS and its upregulation during regeneration in the liver.

### Promoter analysis of NS-regulatory transcription factors (TFs)

To search for potential TFs that might downregulate or upregulate NS expression during liver aging and regeneration, respectively, we used the TRANSFAC® database to dissect TF-binding sites in the NS promoter region (-3.3kb to -1). Multiple binding sites for c-Myc (red bars),

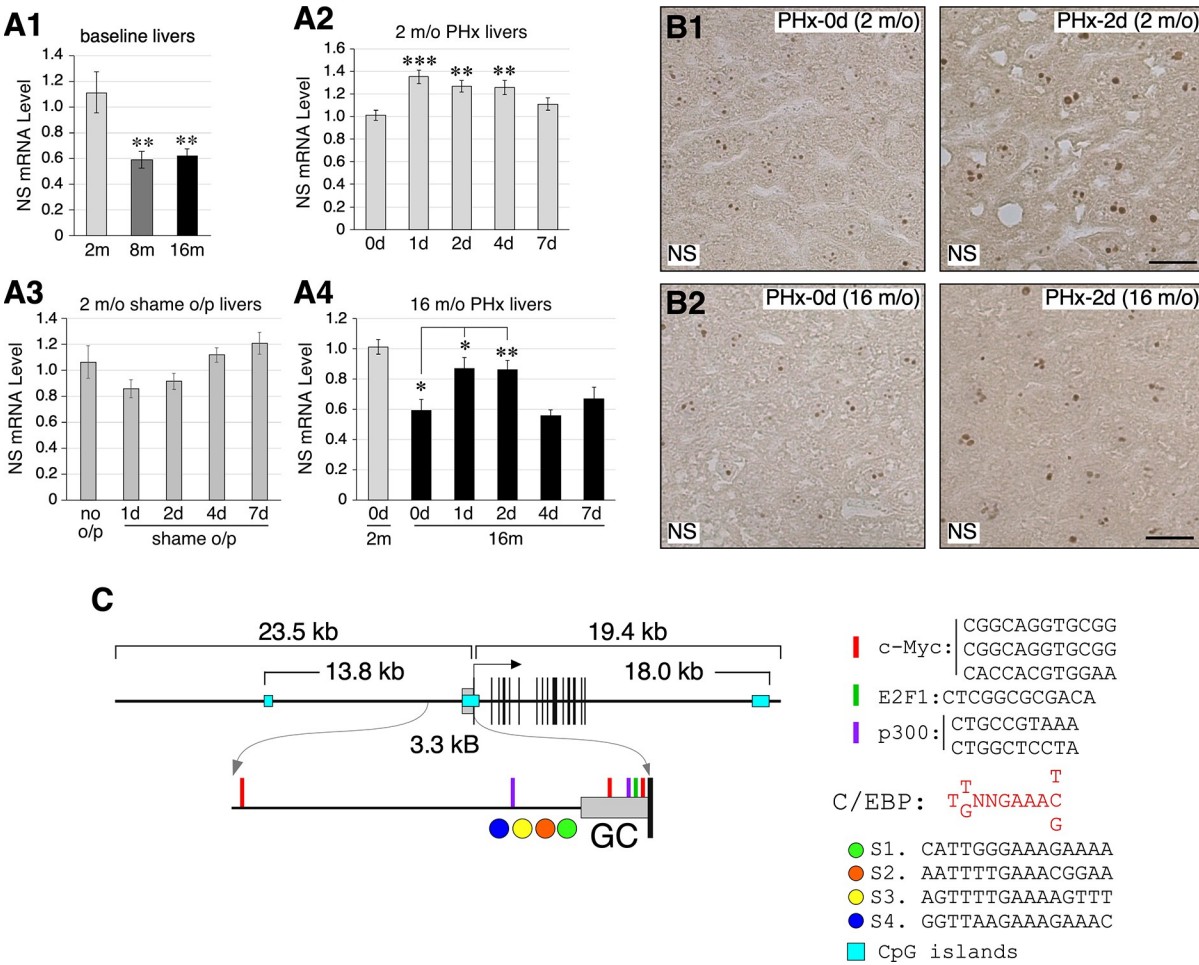

**Fig 1. Expression and promoter analyses of nucleostemin (NS) in aging and regenerating livers.** NS expression by qRT-PCR in (**A1**) young (2 m/o, light gray bar), adult (8 m/o, dark gray bar), and early aged (16 m/o, black bar) livers; (**A2**) 2-m/o livers before (0d) or after 70% partial hepatectomy (PHx) on day 1 (1d), 2d, 4d, or 7d; (**A3**) 2-m/o non-operated livers (no o/p) and sham-operated livers collected after 1d, 2d, 4d, or 7d of the procedure; (**A4**) 16-m/o livers before (0d) or after PHx. The graphs show the results obtained using Rps3 as the internal reference. Bars show mean ± s.e.m. Statistical differences were analyzed by t-test, with p values < 0.05, <0.01, and <0.001 indicated by *, **, and ***, respectively. (**B**) Immunostaining of NS protein signals (brown) in 70% PHx-0d and PHx-2d livers at 2 and 16 m/o. Sections were counterstained with hematoxylin (light blue). Scale bars show 20μm. (**C**) Analysis of transcription factor (TF) binding sites in the NS promoter region (-3.3kb to -1) and CpG islands (CGIs) in the NS genomic locus (-23.5kb to +19.4k). The TRANSFAC search revealed a GC box (-633 to -1) with multiple binding sites for c-Myc (red bars), E2F1 (green bar), and p300 (purple bars) and a CAAT box (-1,216 to -735) with four C/EBPα binding sites (circles). The MethPrimer software identified only one CGI (blue boxes) within 10kb of the NS locus. The black vertical lines represent exons.

E2F1 (green bars), and p300 (purple bars) were found, mostly in a GC box (-633 to -1) (Fig 1B). Upstream of the GC box, a CAAT box with four C/EBPα binding sites (circles) was found (-1216 to -735). c-Myc and E2F1 have been implicated as positive NS regulators in mice and human liver cancer cells, respectively [39,47], whereas C/EBPα and p300 have not yet been linked to NS expression. In addition, MethPrimer analysis identified three CpG islands (blue boxes) within 43kb (-23.5kb to +19.4kb) of the NS transcription start site (TSS), one of which coincided with the GC box.

## Expression profiles of candidate NS-regulatory TFs in liver aging and regeneration

To identify TFs that might regulate NS expression during liver aging and/or regeneration, qRT-PCR assays were performed to match the expression profiles of activating TFs (*e.g.*, c-Myc, E2F1, and p300) (Fig 2) and inhibitory TFs (*e.g.*, C/EBPα and C/EBPβ) (Fig 3) with that of NS (Fig 1A). Our results showed that the expression levels of c-Myc, E2F1, and p300 were reduced in 16-m/o livers compared to those in 2-m/o livers (Fig 2A). During regeneration, 2-m/o livers showed upregulation of c-Myc, E2F1, and p300 expression in different post-PHx time windows, *i.e.*, 2-7d (c-Myc), 1-7d (E2F1), or 4-7d (p300). In 16-m/o livers, the PHx-induced upregulation of c-Myc, E2F1, and p300 was generally dampened, with p300 showing the most significant decrease (Fig 2C). In contrast to these activating TFs, C/EBPα expression remained relatively unchanged (Fig 3A1), and C/EBPβ expression was upregulated during liver aging (Fig 3A2). In response to 70% PHx, 2-m/o livers showed a significant downregulation of C/EBPα expression from 1-7d, with the greatest decrease occurring at 1-2d (Fig 3B1), and a decrease in C/EBPβ at 2d post-PHx (Fig 3B2). In 16-m/o livers, the PHx-elicited expression response of C/EBPα resembled that in 2-m/o livers (Fig 3C1), whereas the decrease in C/EBPβ occurred as early as 1d post-PHx (Fig 3C2). These results show that the time course of NS upregulation during liver regeneration coincides with an increase in E2F1 and a decrease

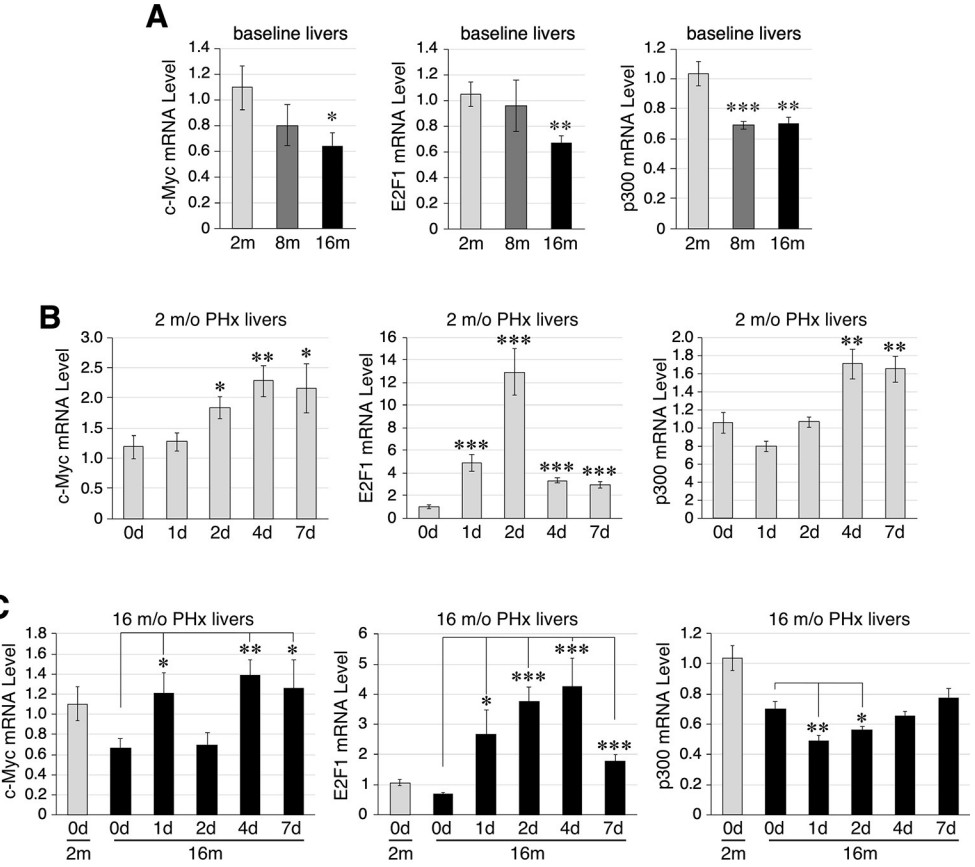

**Fig 2. Expression of E2f1 positively correlates with that of NS in aging and regenerating livers.** qRT-PCR analysis of c-Myc, E2F1, and p300 expression in 2, 8 and 16-m/o uninjured livers (**A**), 2-m/o livers before and after PHx (**B**), and 16-m/o livers before and after PHx (**C**). The graphs show the results in reference to Rps3. See Fig 1 for statistical method and annotations.

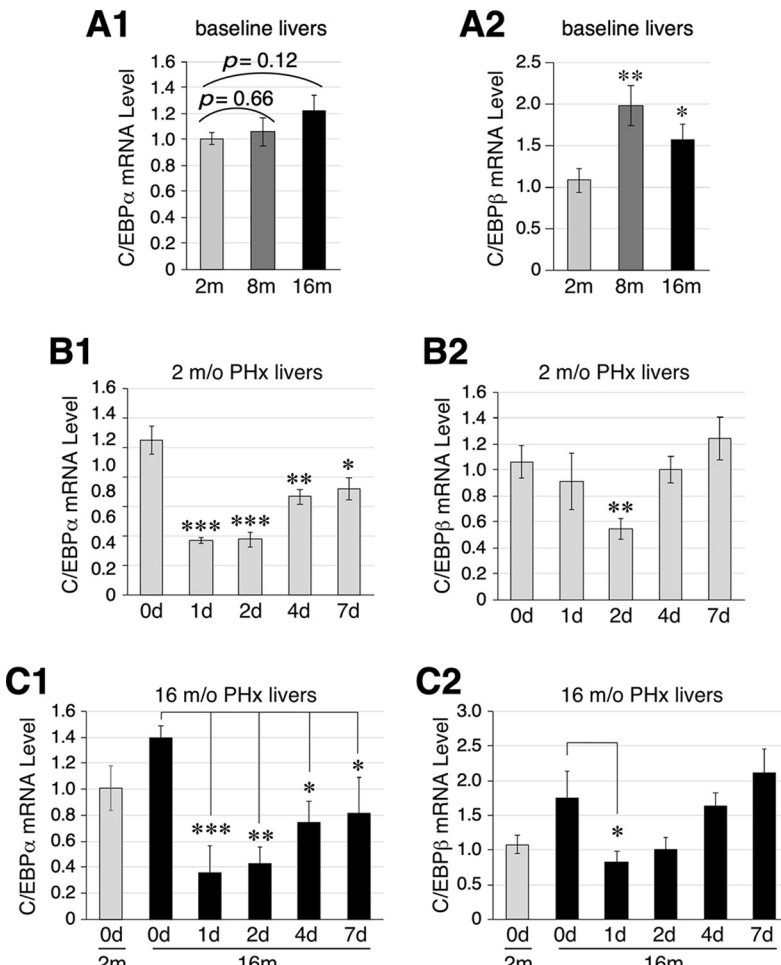

**Fig 3. C/EBPα expression negatively correlates with that of NS in aging and regenerating livers.** qRT-PCR analysis of C/EBPα and C/EBPβ expression in 2, 8, and 16-m/o baseline livers (**A**), 2-m/o livers before and after PHx (**B**), and 16-m/o livers before and after PHx (**C**). The graphs show the results in reference to Rps3. See Fig 1 for statistical method and annotations.

in C/EBPα, both of which occur rapidly within 1d. However, unlike NS or C/EBPα, the level of E2F1 peaks at 2d in 2-m/o livers and 2-4d in 16-m/o livers. In support, Pearson correlation analysis showed that the baseline expression of NS and C/EBPα were inversely correlated in young (2-m/o) and old (16-m/o) livers (r = -0.42, p = 0.052), and the regulated expression of NS and C/EBPα during liver regeneration (D0, D1, D2, D4, and D7) were also inversely correlated at 2 m/o (r = -0.44, p = 0.001) and 16 m/o (r = -0.30, p = 0.038). We therefore conclude that among the five candidate TFs identified by promoter analysis, only C/EBPα reversely matches the temporal expression profiles of NS during liver regeneration, suggesting that it may play a role in suppressing NS expression during liver regeneration.

## C/EBPα binds NS promoter sequences and suppresses its transcription activity

To establish the function of C/EBPα in regulating NS expression, we first determined whether it could directly bind to the C/EBP consensus sequences found in the NS promoter (Fig 1B). Electromobility shift assay (EMSA) was performed using purified C/EBPα protein mixed with

radiolabeled probes with sequences identical to the S1 or S2 site in the NS promoter (NS-S1 and NS-S2) or the C/EBPα-binding motif of PEPCK as a positive control. EMSA confirmed that C/EBPα directly bound the S1 and S2 sequences (Figs 4A and S2). The specificity of the C/EBPα-bound signals was validated by supershift experiments using an anti-C/EBPα anti-body. To confirm that C/EBPα binds the endogenous NS promoter, Chromatin Immunopre-cipitation (ChIP) assays were performed in Hepa1-6 mouse HCC cells with overexpressed mouse C/EBPα protein tagged with the FLAG epitope (FLAG-C/EBPα). The experimental group consisted of samples transfected with FLAG-C/EBPα and pulled down with anti-FLAG antibody. Control groups included: 1) samples transfected with FLAG-C/EBPα and pulled down with IgG, or 2) samples transfected with empty vector and pulled down with anti-FLAG antibody. The amounts of the endogenous NS promoter in the pulldown fractions were mea-sured by qPCR assays using two pairs of primers spanning the C/EBPα binding sites in the NS promoter (Fig 4B1) and expressed as percentages of the input DNA of the same samples. The ChIP results showed that anti-FLAG antibody preferentially pulled down the endogenous NS promoter in Hepa1-6 cells expressing FLAG-C/EBPα (lanes 3 and 6) but not in Hepa1-6 cells without FLAG-C/EBPα (lanes 1 and 4). Nor did IgG pull down the endogenous NS promoter in Hepa1-6 cells expressing FLAG-C/EBPα protein (lanes 2 and 5) (Fig 4B2). All differences were statistically significant ($< 0.05$), except for the comparison between lanes 4 & 6 ($p = 0.08$) due to slightly high backgrounds in anti-FLAG pulldown samples. To determine the role of C/EBPα in regulating NS promoter activity, a luciferase reporter construct was created by sub-cloning the native NS promoter region containing the four C/EBPα–binding sites (×4) into the 3' end of the pGL2-Promoter (SV40) firefly luciferase reporter vector (pGL2-P×4) (Fig 4B1). Hep3B cells were transfected with the firefly reporter plasmid (pGL2-P×4 or pGL2-P), Renilla luciferase reporter plasmid as an internal control, and pCIS expression vector for C/EBPα, C/EBPβ, or vector alone. The promoter activity was determined using a dual-luciferase assay. Our data showed that C/EBPα exerted a strong dose-dependent inhibitory effect on pGL2-P×4 but not on pGL2-P (Fig 4B2). C/EBPβ was found to exert an activation effect on the SV40 promoter of pGL2-P independent of the C/EBP-binding site (Fig 4B3). To correct for this NS-unrelated effect of C/EBPβ on pGL2-P, C/EBPβ-induced and binding site-dependent changes were determined by calculating the ratios of C/EBPβ-transfected versus non-trans-fected samples of the same reporter constructs (lane 3 or 4 divided by lane 2; lane 6 or 7 divided by lane 5) and then comparing pGL2-P×4 vs. pGL2-P transfected samples. We found that C/EBPβ also suppressed transcriptional activity in a C/EBP-binding site-dependent and C/EBPβ dose-dependent manner (Fig 4B4). In comparison, C/EBPα showed stronger inhibi-tory activity than C/EBPβ. To confirm whether C/EBPα or C/EBPβ inhibited endogenous NS expression *in vivo*, we determined the effect of C/EBPα and C/EBPβ on the endogenous expression of NS, c-Myc, E2F1, and p300 in Hep3B cells. Expression analyses showed that C/EBPα overexpression significantly decreased the endogenous expression of NS and, to a lesser extent, the expression of p300, but had no effect on the expression of c-Myc or E2F1 (Fig 4C). In contrast to the luciferase findings, C/EBPβ overexpression did not affect the endogenous expression of NS. Nor did it affect the expression of c-Myc, E2F1, or p300. These findings sug-gest that C/EBPα plays a functional role in downregulating NS expression by directly binding to its promoter.

## NS expression in 2-m/o C/EBPα (S193D) livers mimics that in 16-m/o wildtype livers in regeneration

Mutation of Ser 193 to Asp (S193D) has been shown to mimic the gain-of-function (GOF) effect of C/EBPα phosphorylation. An S193D-C/EBPα knock-in mouse model was created

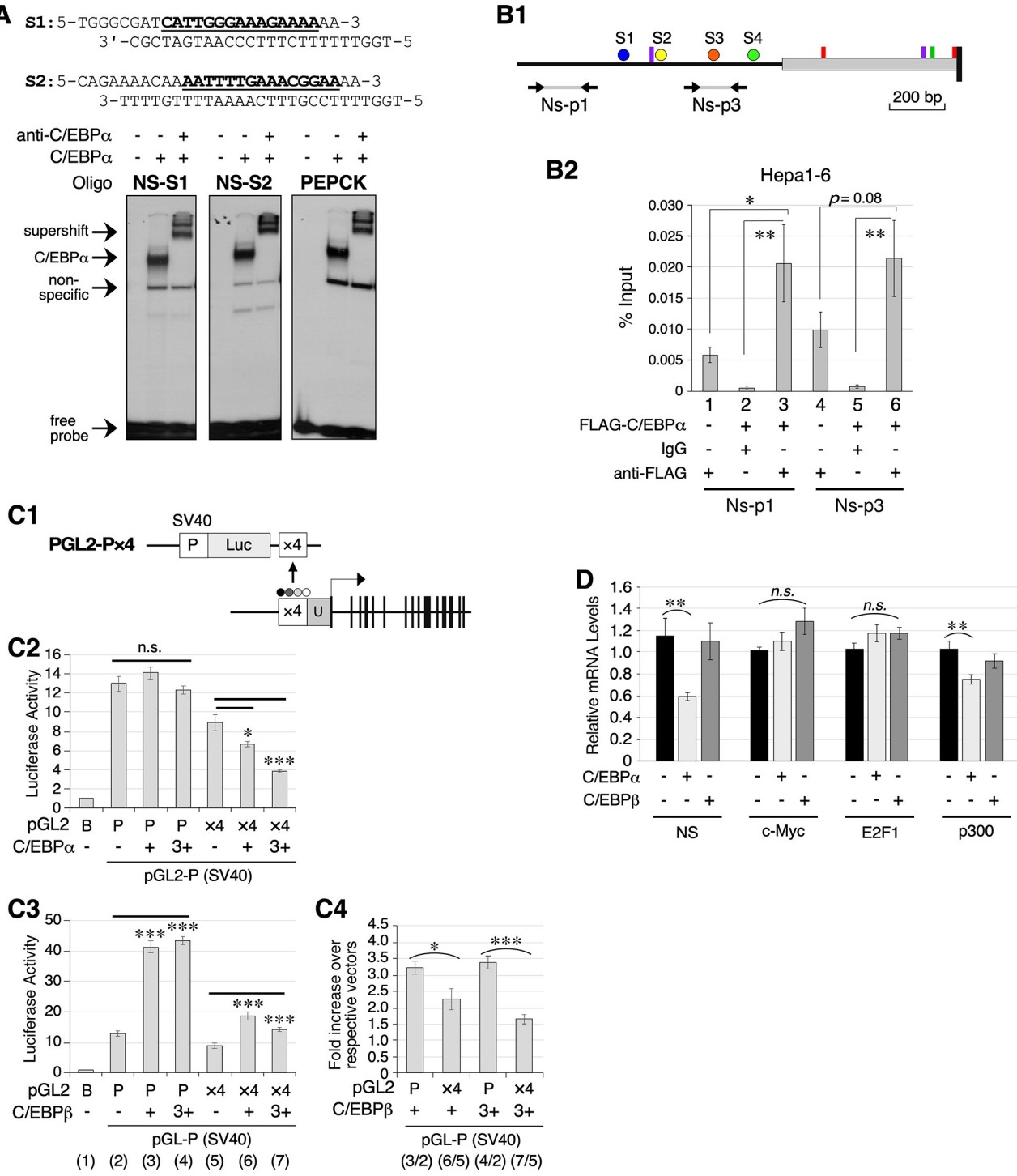

**Fig 4. C/EBPα binds the consensus sequences found in the NS promoter *in vitro* and suppresses its transcriptional activity in Hep3B cells.** (**A**) (Top) Probe sequences corresponding to the S1 and S2 sites found in the endogenous NS promoter. (Bottom) Electromobility shift assays (EMSA) of C/EBPα binding to the NS-S1 and NS-S2 probes and a control probe of C/EBPα-binding sequence in the PEPCK promoter. The specificity of C/EBPα-bound signals was validated by supershift experiments with anti-C/EBPα antibody. (**B**) Chromatin Immunoprecipitation (ChIP) assays on the binding of FLAG-tagged mouse C/EBPα (FLAG-C/EBPα) to the endogenous NS promoter in Hepa1-6 cells. (**B1**) The amounts of the NS promoter in the pulldown fractions were quantified by qPCR using two primer pairs (Ns-p1 and Ns-p3). (**B2**) The experimental group consisted of FLAG-C/EBPα transfected cells pulled down with anti-FLAG antibody (lanes 3 & 6). The control groups consisted of: (1) vector-transfected cells pulled down with anti-FLAG antibody (lanes 1 & 4), or (2) FLAG-C/EBPα-transfected cells pulled down with IgG (lanes 2 & 5). (**C1**) A schematic diagram of a pGL2-P×4 luciferase reporter construct containing a NS promoter region with the four C/EBP-binding sites (×4), subcloned into the 3' end of the

pGL2-Promoter (SV40) luciferase reporter vector. (**C2**) Effects of C/EBPα expression on the transcriptional activity of pGL2-P (P) and pGL2-P×4 (×4) in Hep3B cells were measured by dual-luciferase reporter assays. Results were referenced to the pGL2-B group arbitrarily set as 1. (**C3**) Effects of C/EBPβ on the transcriptional activity of pGL2-P and pGL2-P×4 were measured and referenced to the pGL2-B group. Lane numbers were shown in parenthesis. (**C4**) Effects of C/EBPβ on the transcriptional activity of pGL2-P and pGL2-P×4 were analyzed as ratios to non-C/EBPβ-transfected samples of the same reporter constructs. (**D**) Effects of C/EBPα and C/EBPβ on the expression of endogenous NS, c-MYC, E2F1, and p300 in Hep3B cells in reference to RPS3. See Fig 1 for statistical method and annotations.

that showed premature liver aging phenotypes [48,49]. To demonstrate the C/EBPα function in regulating NS expression in mouse livers, we compared NS expression between young (2-m/o) S193D and wild-type livers. Under the baseline condition, the NS expression level of 2-m/o S193D livers was not significantly different from that of 2-m/o wild-type livers (Fig 5A1). During PHx-induced regeneration, the upregulation of NS in S193D livers resembled that of 16-m/o livers in its early return to the baseline level at 4d but differed from 16-m/o livers in showing a biphasic increase at 7d post-PHx (Fig 5A2). Under the baseline condition, only p300, but not c-Myc or E2F1, showed a decrease in 2-m/o S193D livers compared to 2-m/o wild-type livers (Fig 5B1, 5C1 and 5D1). During PHx-induced regeneration, upregulation of c-Myc, E2F1, and p300 was diminished in S193D livers compared to 2-m/o wild-type livers, with the exception of the c-Myc level at 7d post-PHx (Fig 5B2, 5C2 and 5D2). These results show that C/EBPα GOF by the S193D mutation dampens the upregulation of NS in regenerating livers without affecting the baseline expression of NS in aging livers.

## C/EBPα knockdown (KD) increases NS expression in aged livers

Our findings indicate a functional role for C/EBPα in suppressing NS expression during liver regeneration. To test this hypothesis *in vivo*, we determined the effects of C/EBPα knockdown

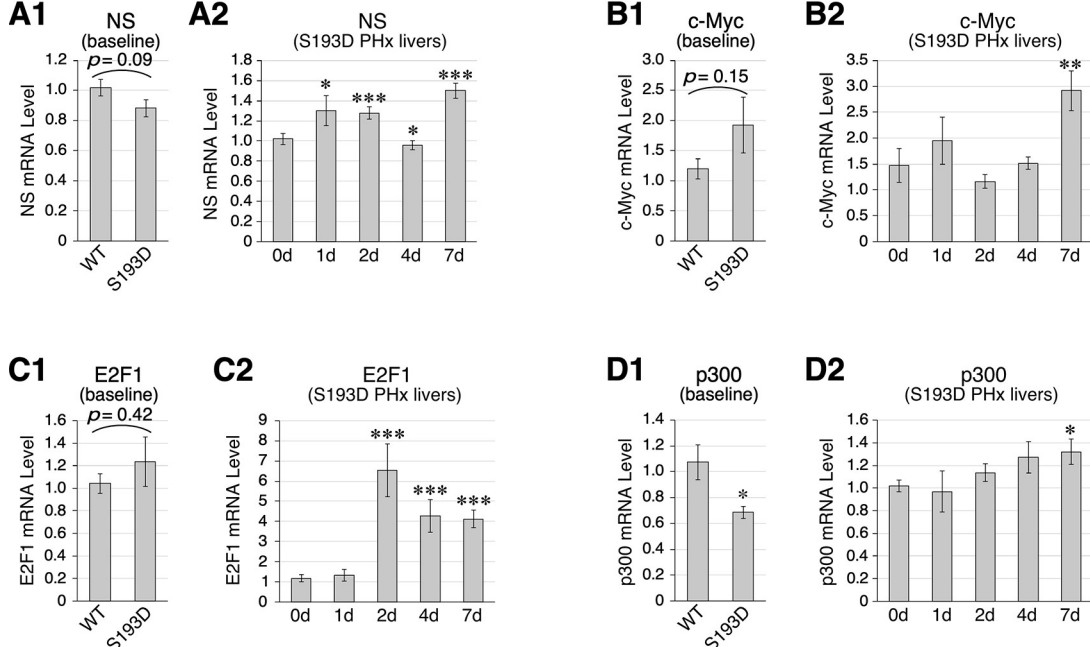

**Fig 5. Effects of gain-of-function mutation of C/EBPα (S193D) on NS, c-Myc, E2f1, and p300 expression in 2-m/o livers under the baseline and regeneration conditions.** (**A**) qRT-PCR assays of NS expression in 2-m/o S193D-C/EBPα livers at baseline, compared to wildtype (WT) livers (**A1**), or during PHx-induced regeneration (**A2**). Same studies were performed for c-Myc (**B**), E2f1 (**C**), and p300 (**D**). Graphs show results in reference to Rps3. See Fig 1 for statistical method and annotations.

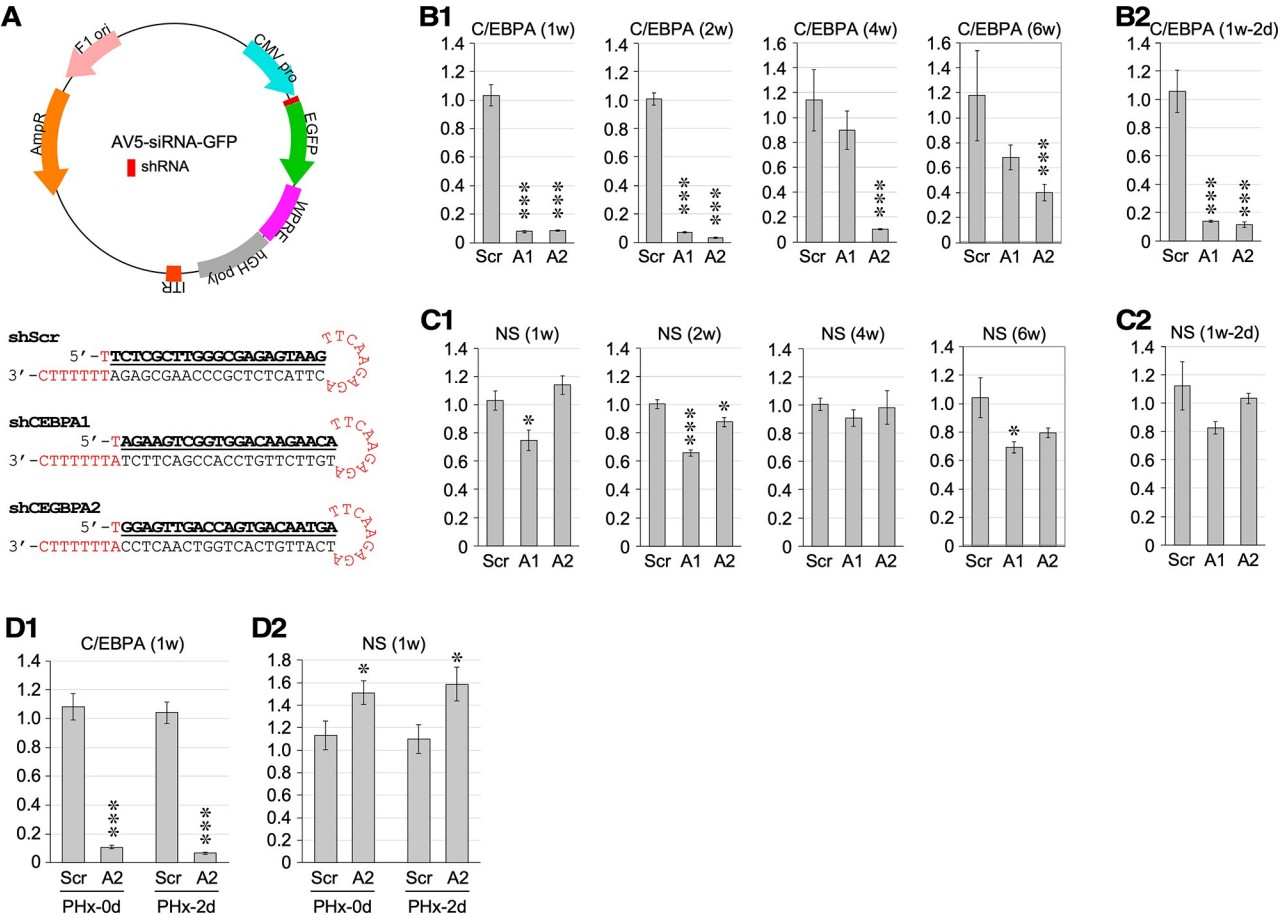

**Fig 6. C/EBPα depletion increases NS expression in 24-m/o livers but not in 2-m/o livers.** (**A**) (Top) Diagram of AV5-siRNA-GFP plasmid containing a CMV promoter-driven shRNA sequence, designed to make adeno-associated virus serotype 8 (AAV8) for *in vivo* knockdown experiments. (Bottom) Short hairpin RNAi constructs designed to target 21-nucleotide sequences on C/EBPα (A1 and A2) or a scrambled 21-nucleotide sequence (Scr). (**B**, **C**) Hepatic expression of C/EBPα (**B1**) and NS (**C1**) in 2-m/o mice injected with AAV8-shRNA (1E+11 gc in 50μL) and harvested after 1 week (1w), 2w, 4w, and 6w. Hepatic expression of C/EBPα (**B2**) and NS (**C2**) in 2-m/o mice injected with AAV8-shRNA, receiving 70% PHx one week after AAV injection, and sacrificed 2 days after PHx. Hepatic expression of C/EBPα (**D1**) and NS (**D2**) in 24-m/o mice injected with AAV8-shRNA (1E+11 gc in 50μL) and sacrificed after 1w before (PHx-0d) or after 1/3 PHx 2d later (PHx-2d). Graphs show results in reference to Rps3. See Fig 1 for statistical method and annotations.

on NS expression in the liver under baseline and regenerative conditions. Two C/EBPα-targeting shRNAs and one control shRNA were created and packaged into adeno-associated virus (AAV) serotype 8 (AAV8) for *in vivo* delivery (Fig 6A). AAV8 is a robust vector for gene delivery to the liver with a minimal tissue immune response. To introduce C/EBPα-KD, 2-m/o mice were injected with AAV8-Scr (control), AAV8-C/EBPA1 (C/EBPα-KD construct 1), or AAV8-C/EBPA2 (C/EBPα-KD construct 2) at 1E+11 genome copies (gc) per mouse via portal vein injection. To evaluate KD efficiency, liver samples were collected and analyzed for C/EBPα expression. qRT-PCR assays showed that C/EBPα expression in the liver was significantly depleted by both AAV8-C/EBPA1 and AAV8-C/EBPA2 constructs from one week (1w) to 4 w post-injection, with peak KD effect observed between 1-2w (Figs 6B1 and S3A), and the KD effect was also seen at 1d post-PHx (Fig 6B2). AAV8-C/EBPA2 showed a stronger and longer KD effect than AAV8-C/EBPA1. At 2 m/o, little or no change in NS expression was seen in C/EBPA2-knockdown livers and a decrease in NS expression was seen in C/EBPA1-knockdown livers under the baseline condition from 1 to 6 weeks after AAV injection (Figs 6C1 and

S3B), and under the PHx-induced regenerative condition, no difference in NS expression was observed between control-knockdown and C/EBPA1- or C/EBPA2-knockdown livers at 2d after the surgery (Fig 6C2).

The lack of changes in NS expression in C/EBPα-KD livers by C/EBPA2 under baseline and regenerative conditions suggests that the C/EBPα-mediated suppression mechanism of NS expression might be active only at old age. To test this hypothesis, 20-24-m/o mice were injected with AAV8-C/EBPA2 via the portal vein and received 1/3 PHx one week later. Here, we focused on the effect of AAV8-C/EBPA2, because AAV8-C/EBPA1 displayed a relatively short-lived knockdown effect and elicited a paradoxical decrease in NS expression in 2-m/o livers. We first confirmed that AAV8-C/EBPα injection significantly decreased C/EBPα expression in aged livers under baseline and regeneration conditions (Fig 6D1). Notably, NS expression significantly increased following C/EBPα-KD in aged livers under both baseline and regeneration conditions (Fig 6D2). These results support the C/EBPα suppression of NS expression in aged livers but not in young livers.

## Discussion

Adult livers possess the unique capability to regain their original size after removing a large chunk of their mass. This mythological feature of the liver employs a dual mechanism that either rekindles mitotically quiescent mature hepatocytes to reenter the cell cycle or, under extensive damage conditions, activates hepatic stem cells to regenerate both hepatocytes and bile duct epithelial cells. The regenerative capability of the liver is considerably compromised with age, resulting in multiple health-related issues. NS plays a critical role in stem cell self-renewal and liver regeneration by protecting the genome from replicative DNA damage. In this study, we report new findings regarding the transcriptional regulation of NS expression during liver aging and regeneration, providing a new perspective on how tissue homeostasis is orchestrated during the aging process.

### Changes in NS expression inversely match that in C/EBPα during liver regeneration

Aging attenuated the baseline expression of NS and its overall expression during liver regeneration The dynamics of NS expression during regeneration in 16-m/o livers resembled those in 2-m/o livers, except for an early return to the preoperative level (4d). To dissect the transcriptional mechanism underlying the age-dependent regulation of NS, five TFs were identified based on their consensus binding sites in the NS promoter. Among the genes differentially expressed during liver regeneration, NS showed a notable early onset response within 1d, consistent with its role in protecting hepatocytes from replication-induced DNA damage. Therefore, we reasoned that *bona fide* NS-regulatory TFs would display a temporal expression profile that either matched or preceded the expression of NS during liver regeneration. Of the five candidate TFs, only E2f1 and C/EBPα exhibited early changes within 1d, and only C/EBPα showed peak changes at 1d and 2d, matching that of NS, whereas E2f1 displayed its peak expression level at 2d. Upregulation of c-Myc and p300 began at 2d and 4d, respectively, and downregulation of C/EBPβ began at 2d post-PHx.

### C/EBPα-mediated direct and indirect suppression of NS expression

C/EBPα plays a master role in orchestrating the expression of hepatocyte proliferative factors involved in liver regeneration in an age-dependent manner. Our data showed that C/EBPα bound directly to the C/EBP-binding consensus sequences found in the NS promoter region *in vitro*. Functional perturbation studies demonstrated that C/EBPα attenuated the

transcriptional activity of the NS promoter using four experimental paradigms, including cell-based luciferase assay, endogenous NS expression in Hep3B cells, mouse livers with a GOF mutation of C/EBPα (S193D), and mouse livers with C/EBPα knockdown. S193D mimics the phosphorylation status of C/EBPα, which increases with age via a cyclin D3- and GSK3β-mediated mechanism [50]. At a young age (2 m/o), S193D livers differed from 2-m/o wild-type livers in their NS expression profiles during PHx-induced regeneration, showing an early return to the baseline level at 4d and a biphasic increase at 7d. Interestingly, this feature of early return to the baseline level at 4d was also noted in the 16-m/o livers. Conversely, C/EBPα depletion increased the expression of NS under both baseline and regenerative (PHx-2d) conditions in aged livers (24 m/o), but not in young livers (2 m/o). These findings support the hypothesis that C/EBPα-mediated inhibition of NS expression affects aged livers. Young mice have a higher regenerative capability and more NS expression in their livers compared to aged mice. In conjunction with the effect of C/EBPα knockdown on NS expression in aged mice but not in young mice, we hypothesize that the C/EBPα-mediated inhibition on NS transcription occurs mainly at the old age and may be controlled by post-translational modification such as phosphorylation at S193.

In addition to the direct regulation of NS expression, C/EBPα phosphorylation also promotes its complexation with Brm (Brahma chromatin remodeling complex) or HDAC1, which inhibits the expression of c-Myc, b-Myb, Cdc2, and FoxM1B after PHx or the expression of c-Myc and FOXM1B, respectively [6,19,51]. Given that c-Myc has been shown to regulate NS expression [39,47], it is conceivable that C/EBPα may also downregulate NS via an indirect mechanism of c-Myc inhibition. Furthermore, the C/EBPα-Brm complex downregulates E2F-dependent genes [20], and E2F-regulated genes are highly co-enriched with NS in HCC samples [39], raising another indirect mechanism by which C/EBPα might regulate NS via E2F suppression. Our results showed that: (1) changes in C/EBPα and NS expression during liver regeneration coincided with each other and preceded that of c-Myc and E2F1, and (2) NS C/EBPα overexpression significantly downregulated NS without changing the expression of c-Myc or E2F1 in Hep3B cells. These findings support a direct role for C/EBPα in regulating NS expression instead of an indirect role via c-Myc and/or E2F1 regulation. Nevertheless, we cannot completely rule out the possibility that c-Myc and/or E2F1 contribute to NS regulation in a C/EBPα-independent manner at later time points. Aging also increases C/EBPβ expression. Age-dependent increases in the C/EBPβ-HDAC1 complex are involved in the repression of GSK3β and SIRT1 promoters, causing impaired homeostasis and liver proliferation [8,52]. In our experimental systems, even though C/EBPβ displayed a less evident inhibitory effect on NS expression in the cell-based reporter assay, it did not affect endogenous NS expression in Hep3B cells, and the dynamic profile of C/EBPβ expression in liver regeneration did not match that of NS.

In conclusion, this study reports the discovery of C/EBP-alpha as an upstream regulator of NS that functions by negatively controlling the NS expression in aging livers. Understanding how NS is transcriptionally regulated during the regeneration and aging of liver and in response to different environmental factors (e.g., circadian oscillation, nutrition, etc.) via the C/EBPα pathway may provide a broad perspective on the mechanistic orchestration of tissue homeostasis in the aging process.

## Supporting information

**S1 Fig. Liver-to-body weight indices during liver regeneration at 2 and 16 months old.**
(PDF)

**S2 Fig. Uncropped and unadjusted images of EMSA data shown in Fig 4A.**
(PDF)

**S3 Fig. Hepatic expression of C/EBPα and NS in 2-m/o mice injected with AAV8-shRNA.**
(PDF)

## Acknowledgments

We thank Yue Chen, Wenrui Ye, and Britni Trinh for their technical support, Baylor College of Medicine Neurovisualization Core (Dr. Tao Lin) for performing anti-NS IHC, and Dr. Stefan Siwko for reviewing our manuscript before submission.

## Author Contributions

**Conceptualization:** Nikolai Timchenko, Robert Y. L. Tsai.

**Data curation:** Robert Y. L. Tsai.

**Formal analysis:** Xiaoqin Liu, Junying Wang, Nikolai Timchenko, Robert Y. L. Tsai.

**Funding acquisition:** Robert Y. L. Tsai.

**Investigation:** Xiaoqin Liu, Junying Wang, Fang Li, Nikolai Timchenko, Robert Y. L. Tsai.

**Project administration:** Robert Y. L. Tsai.

**Supervision:** Robert Y. L. Tsai.

**Visualization:** Robert Y. L. Tsai.

**Writing – original draft:** Robert Y. L. Tsai.

**Writing – review & editing:** Xiaoqin Liu, Junying Wang, Fang Li, Nikolai Timchenko, Robert Y. L. Tsai.

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
