## [Decision Letter · Decision Letter 0]

16 Jun 2024

PONE-D-24-19975Transcriptional Control of A Stem Cell Factor Nucleostemin in Liver Regeneration and AgingPLOS ONE

Dear Dr. Tsai,

Thank you for submitting your manuscript to PLOS ONE. After careful consideration, we feel that it has merit but does not fully meet PLOS ONE’s publication criteria as it currently stands. Therefore, we invite you to submit a revised version of the manuscript that addresses the points raised by the reviewer.

We look forward to receiving your revised manuscript.

Kind regards,

Matias A Avila, Ph.D.

Academic Editor

PLOS ONE

Journal Requirements:

2. To comply with PLOS ONE submissions requirements, in your Methods section, please provide additional information regarding the experiments involving animals and ensure you have included details on methods of sacrifice.

"This work was supported by NIA-PHS grant R21AG052006 and the Cancer Prevention Research Institute of Texas (CPRIT) Individual Investigator Research Award (RP200081) to RYT. "

4. Please expand the acronym “NIA-PHS” (as indicated in your financial disclosure) so that it states the name of your funders in full.

"We thank Yue Chen, Wenrui Ye, and Britni Trinh for their technical support, and Dr. Stefan Siwko for reviewing our manuscript before submission. This work was supported by NIA-PHS grant R21AG052006 and the Cancer Prevention Research Institute of Texas (CPRIT) Individual Investigator Research Award (RP200081) to RYT. There is no conflicts of interest to report. "

"This work was supported by NIA-PHS grant R21AG052006 and the Cancer Prevention Research Institute of Texas (CPRIT) Individual Investigator Research Award (RP200081) to RYT. "

6. Please provide a complete Data Availability Statement in the submission form, ensuring you include all necessary access information or a reason for why you are unable to make your data freely accessible. If your research concerns only data provided within your submission, please write "All data are in the manuscript and/or supporting information files" as your Data Availability Statement.

7. PLOS ONE now requires that authors provide the original uncropped and unadjusted images underlying all blot or gel results reported in a submission’s figures or Supporting Information files. This policy and the journal’s other requirements for blot/gel reporting and figure preparation are described in detail at https://journals.plos.org/plosone/s/figures#loc-blot-and-gel-reporting-requirements and https://journals.plos.org/plosone/s/figures#loc-preparing-figures-from-image-files. When you submit your revised manuscript, please ensure that your figures adhere fully to these guidelines and provide the original underlying images for all blot or gel data reported in your submission. See the following link for instructions on providing the original image data: https://journals.plos.org/plosone/s/figures#loc-original-images-for-blots-and-gels.   

Reviewers' comments:

Reviewer's Responses to Questions

**Comments to the Author**

1. Is the manuscript technically sound, and do the data support the conclusions?

Reviewer #1: Partly

2. Has the statistical analysis been performed appropriately and rigorously? 

Reviewer #1: I Don't Know

3. Have the authors made all data underlying the findings in their manuscript fully available?

Reviewer #1: Yes

4. Is the manuscript presented in an intelligible fashion and written in standard English?

Reviewer #1: Yes

5. Review Comments to the Author

Reviewer #1: This article aims to elucidate the transcriptional mechanisms of nucleostemin (NS) expression during liver regeneration during both liver regeneration in aged livers. The study shows that C/EBPα binding to the NS promoter suppresses its transcriptional activity, leading to reduced NS in this context. In order for this manuscript to be accepted, it is necessary to conduct additional experiments and/or determinations, and clarify certain aspects of the results and discussion, along with correcting certain formal aspects

1. Sham-operated animals should be included to analyze whether the variation in NS levels could be due to the acute-phase response of the laparotomy.

2. Did you observe a delay in the liver index in elderly mice at the end of the experiment? These data should be shown or discussed.

3. What is the significance of NS increase in liver regeneration? You should provide any determination of the impact of defective NS regulation on liver regeneration in elderly mice.

4. It is known that C/EBPα levels change with the circadian rhythm. How could this oscillation affect NS expression? The nutritional status of the mice during the experiment should be explained in the methods.

5. Although basal NS mRNA levels are lower in adult and aged livers, the elevation following partial hepatectomy resembles that of young livers. This suggests that the regulatory mechanisms are similar. It is essential to analyze NS protein levels at selected time points after partial hepatectomy (PH). Do NS and C/EBP (inversely) correlate when performing Pearson correlation analysis?

6. Although EMSA has already demonstrated that C/EBPα binds to the consensus sequences in the NS promoter in vitro, conducting a ChIP would allow verification of this interaction within the context of a chromosome in living cells.

7. Why do AAVs have no effect on young mice? Why did you use 24-month-old mice in AAV studies? Why do you perform a 30% hepatectomy and not a 70% hepatectomy? You need to hypothesize why AAV does not work in young mice but does in old mice.

8. The statistical methods are not described.

Formal aspects:

1. The title "aging livers" in Figure 2 is inappropriate. In other graphs, the titles do not correctly inform what is being represented.

2. Graphs in Figure 5 would be clearer if the gene titles were included and the basal data were combined with the hepatectomized liver data in the same graph.

3. Graphs 6B1 and 6C1 would be better understood as line graphs (time series).

4. In some graphs, the curved lines indicating significance are not used correctly. Review and standardize their use.

5. Please correct "C/ebp alpha" written in lowercase. Standardize its writing.

6. Some points are missing (at the end of sentences).

6. PLOS authors have the option to publish the peer review history of their article (what does this mean?). If published, this will include your full peer review and any attached files.

Reviewer #1: No

---

## [Author Response · Author response to Decision Letter 0]

16 Aug 2024

We sincerely thank the academic editor and the reviewer for their valid and constructive comments on our manuscript from many angles. We have addressed all the raised critiques by providing new data (Figs. 1A3, 1B, 4B, and S1), new analyses, better clarifications, and justifications (all marked in red in the manuscript). Our point-to-point responses are listed as follows.

Academic editor

1. Make sure our manuscript meets PLOS ONE's style requirements: checked

2. In the Methods section, provide additional information regarding the experiments involving animals and ensure you have included details on methods of sacrifice: provided

3. Please include this amended Role of Funder statement in your cover letter: included

4. expand the acronym “NIA-PHS”: done

5. Update your Funding Statement: The funding statement of two funders (NIA-PHS and CPRIT) in the manuscript is correct.

6. Complete Data Availability Statement in the submission form: A Data Availability Statement is provided in the manuscript.

7. Upload original uncropped and unadjusted images: EMSA images were included in the Supplemental Information (Fig. S2).

Reviewer

1. Sham-operated animals should be included to analyze whether the variation in NS levels could be due to the acute-phase response of the laparotomy.

Response: We have performed sham-operation on 2-month-old mice to determine the NS response to the laparotomy without PHx. The result, now shown in Fig. 1A3, demonstrates that the upregulation of NS is not caused by a response to laparotomy. NS increase during liver regeneration is consistent with previous findings (PMID: 22775537 and 23813570).

2. Did you observe a delay in the liver index in elderly mice at the end of the experiment? These data should be shown or discussed. 

Response: Liver-to-Body Weight Ratios at 1, 2, 4, and 7 days after 70% PHx at 2 and 16 m/o are presented in Fig. S1.

3. What is the significance of NS increase in liver regeneration? You should provide any determination of the impact of defective NS regulation on liver regeneration in elderly mice.

Response: Determination of the impact of defective NS regulation on liver regeneration is interesting but beyond the scope of this study. The focus of this study is to investigate the transcriptional control of NS. The impact of NS loss on liver regeneration was examined using an Albumin-Cre driven NS-knockout mouse model in CCl4-induced liver regeneration (PMID: 23813570), which was cited in the Introduction.

4. It is known that C/EBPα levels change with the circadian rhythm. How could this oscillation affect NS expression? The nutritional status of the mice during the experiment should be explained in the methods.

Response: While the effect of many environmental factors (e.g., circadian rhythm, nutrition, etc) in regulating NS expression and hence affecting liver regeneration and aging is potentially interesting but beyond the scope of this study. We appreciate the comment and include this idea in the conclusion statement. We also include the following statement in the Method section: Mice were housed in a vivarium with the light-dark cycle set at 12h-12h with lights on at 06:00 and off at 18:00, and allowed free access to normal chow diet and water.

5. Although basal NS mRNA levels are lower in adult and aged livers, the elevation following partial hepatectomy resembles that of young livers. This suggests that the regulatory mechanisms are similar. It is essential to analyze NS protein levels at selected time points after partial hepatectomy (PH). Do NS and C/EBP (inversely) correlate when performing Pearson correlation analysis?

Response: Anti-NS IHC results at selected time points after PHx were provided in Fig. 1B. Pearson correlation analysis showed that the baseline expression of NS and C/EBPα were inversely correlated in young (2-m/o) and old (16-m/o) livers (r= -0.42, p = 0.052), and the regulated expression of NS and C/EBPα during liver regeneration (D0, D1, D2, D4, and D7) were also inversely correlated at 2 m/o (r= -0.44, p = 0.001) and 16 m/o (r= -0.30, p = 0.038).

6. Although EMSA has already demonstrated that C/EBPα binds to the consensus sequences in the NS promoter in vitro, conducting a ChIP would allow verification of this interaction within the context of a chromosome in living cells.

Response: ChIP assays were performed in living Hepa1-6 cells (mouse HCC cells) to confirm the binding of FLAG-tagged mouse C/EBPα protein to the endogenous NS promoter. The results are presented in Fig. 4B. We used FLAG-tagged mouse C/EBPα for ChIP because we were unable to ChIP the endogenous C/EBPα with the anti-C/EBPα antibody even on the DGAT2 promoter (a positive control). 

7. Why do AAVs have no effect on young mice? Why did you use 24-month-old mice in AAV studies? Why do you perform a 30% hepatectomy and not a 70% hepatectomy? You need to hypothesize why AAV does not work in young mice but does in old mice.

Response: Young mice have a higher regenerative capability and more NS expression in their livers compared to aged mice. In conjunction with the effect of C/EBP� knockdown on NS expression in aged mice but not in young mice, we hypothesize that the C/EBP�-mediated inhibition on NS transcription occurs mainly at the old age and may be controlled by post-translational modification such as phosphorylation at S193 (included in Discussion). We perform 30% instead of 70% PHx in 24-month mice because the mortality rate of 70% PHx is much higher than 30% PHx at the old age (included in Methods).

8. The statistical methods are not described.

Response: Statistical methods were added to the Methods as well as in the Figure legends. 

9. The title "aging livers" in Figure 2 is inappropriate. In other graphs, the titles do not correctly inform what is being represented.

Response: Titles of “aging livers” were revised to “baseline livers”, although our original usage of the term “aging livers” is to refer to livers in the aging (ongoing) process from 2-to-8-to-16 m/o, not “aged” livers.

10. Graphs in Figure 5 would be clearer if the gene titles were included and the basal data were combined with the hepatectomized liver data in the same graph.

Response: Gene symbols were included in the titles. We appreciate the reviewer’s suggestion of combining basal w/ PHx data, but since these two sets of studies were not performed in parallel, we think it is prudent not to combine them.

11. Graphs 6B1 and 6C1 would be better understood as line graphs (time series).

Response: Line graphs for 6B1 and 6C1 are provided in Fig. S3.

12. In some graphs, the curved lines indicating significance are not used correctly. Review and standardize their use.

Response: non-significant p values were provided for all curved lines.

13. Please correct "C/ebp alpha" written in lowercase. Standardize its writing.

Response: C/EBP was used throughout the ms based on the suggestion. In the previous version, we used the lower case to refer to mouse genes and the upper case to refer to human genes.

14. Some points are missing (at the end of sentences).

Response: Assuming point means period, we’ve added periods to all the figure legend titles. Please let us know if we miss any additional ones.

---

## [Editor Report · Decision Letter 1]

27 Aug 2024

Transcriptional Control of A Stem Cell Factor Nucleostemin in Liver Regeneration and Aging

PONE-D-24-19975R1

Dear Dr. Tsai,

We’re pleased to inform you that your manuscript has been judged scientifically suitable for publication and will be formally accepted for publication once it meets all outstanding technical requirements.

Kind regards,

Matias A Avila, Ph.D.

Academic Editor

PLOS ONE
---

## [Editor Report · Acceptance letter]

3 Sep 2024

PONE-D-24-19975R1 

PLOS ONE

Dear Dr. Tsai, 

I'm pleased to inform you that your manuscript has been deemed suitable for publication in PLOS ONE. Congratulations! Your manuscript is now being handed over to our production team.

Kind regards, 

on behalf of

Dr Matias A Avila 

Academic Editor

PLOS ONE